# Prevalence of *Toxoplasma gondii* Antibodies in Individuals Occupationally Exposed to Livestock in Portugal

**DOI:** 10.3390/pathogens11050603

**Published:** 2022-05-22

**Authors:** Daniela Almeida, João Quirino, Pedro Matos, Fernando Esteves, Rita Cruz, Helena Vala, João R. Mesquita

**Affiliations:** 1ICBAS—School of Medicine and Biomedical Sciences, Porto University, Rua de Jorge Viterbo Ferreira, 228, 4050-313 Porto, Portugal; 2Department of Dermatology and Venereology, Centro Hospitalar Universitário São João EPE, Alameda Prof. Hernâni Monteiro, 4200-319 Porto, Portugal; pedro.rolo.matos@chsj.min-saude.pt; 3Instituto Politécnico de Viseu, Campus Politécnico, Escola Superior Agrária de Viseu, 3504-510 Viseu, Portugal; festeves@esav.iv.pt (F.E.); rcpaiva@esav.ipv.pt (R.C.); hvala@esav.ipv.pt (H.V.); 4CERNAS, Instituto Politécnico de Viseu, Campus Politécnico, Escola Superior Agrária de Viseu, 3504-510 Viseu, Portugal; 5Epidemiology Research Unit (EPIUnit), Instituto de Saúde Pública da Universidade do Porto, 4050-313 Porto, Portugal; 6Laboratório para a Investigação Integrativa e Translacional em Saúde Populacional (ITR), 4050-313 Porto, Portugal; 7Centre for the Research and Technology of Agro-Environmental and Biological Sciences (CITAB), University of Trás-os-Montes e Alto Douro, 5001-801 Vila Real, Portugal

**Keywords:** *Toxoplasma gondii*, occupational exposure, antibodies, one health

## Abstract

Toxoplasmosis is a worldwide zoonotic disease with different and complex routes for transmission. Workers occupationally exposed to animals or raw meat and viscera (WOE) may be at more risk than the general population, however conflicting data exist on the risk of developing toxoplasmosis due to this close contact. To add knowledge to this topic, the aim of the present study was to ascertain if WOE were more likely to be anti-*T. gondii* IgG seropositive than the general population as well as to study risk factors for *T. gondii* infection such as professional activity, gender, age, years of work and region. For this purpose, a case–control study using archived samples was setup. A total of 114 WOE (including pig slaughterhouse workers, butchers, veterinarians and farmers) and 228 anonymous volunteers (matched with cases by region, age and gender) were studied for anti-*T. gondii* IgG. A significantly higher anti-*T. gondii* IgG occurrence (*p* = 0.0282) was found in WOE when compared with the general population (72.8% [CI = 64.6–81.0%] versus 60.1% [CI = 54.6–65.6%]). Multivariate analysis showed that WOE of more than 50 years of age were more likely to be seropositive for anti-*T. gondii* IgG (aOR = 16.8; 95% CI 3.6–77.5; *p* < 0.001) than those aged less than 50 years. To our knowledge, this is the first case–control study on the prevalence of anti-*T. gondii* IgG in WOE in Portugal, also showing an added risk for *T. gondii* infection in those exposed to animals or their meat and viscera.

## 1. Introduction

Toxoplasmosis is considered to be one of the most important parasitic zoonoses worldwide [1]. *Toxoplasma gondii* is a facultative heteroxenous protozoan with a complex life cycle and multiple transmission routes, capable of infecting, hypothetically, all warm-blooded animals, including humans [2,3]. Although infection by *T. gondii* is typically asymptomatic, it might cause life-threatening disease in the immunosuppressed, and abortion/congenital disease of the foetus in pregnant women infected by the parasite for the first time during pregnancy [4]. These facts justify the need to determine *T. gondii* occurrence not only in humans but also in animals that act as reservoirs. In domestic animals, congenital toxoplasmosis affects especially small ruminants (essentially sheep) and pigs, causing abortion and congenital malformation of foetuses [4,5,6].

In the last few years, determining risk factors for *T. gondii* infection in humans has been a priority for investigators. This allowed identification of contact with soil, age, poor hygiene, consumption of raw and undercooked meat, drinking of untreated water, consumption of uncleaned vegetables and contact with animals [1,7,8,9] as some of the most important risk factors. A few epidemiological serosurveys tried to ascertain the risk for T. gondii infection in individuals in close contact with animals and their raw meat and viscera [10,11]. Despite this, the majority of these surveys are descriptive and only a few case–control studies have been developed [10,12]. Nevertheless, different professional activities have been suggested to be particularly at risk of *T. gondii* infection. For example, slaughterhouse workers, who can inadvertently ingest raw meat and be contaminated via trophozoites through wounds, when no protection materials are used, or contact with sporulated oocysts in animal fur during flaying, and veterinarians, because of contact with cats, as well as small ruminant placenta, or performing necropsy without protection [2,8,9,13,14,15,16,17]. As such, the risk of exposure is dependent on the type of animal, the hygiene of the workers and the function they perform (evisceration, flaying, carcass cutting) [10,16,18,19].

To increase the knowledge of this important zoonosis, we aimed to ascertain if workers occupationally exposed to animals or raw meat and viscera (WOE) were more prone to *T. gondii* seropositivity than controls, by developing a case–control study on anti-*T. gondii* IgG in WOE and matched general-population controls.

## 2. Results

From the 114 WOE sera samples tested, 83 are positive for the presence of anti-*T. gondii* IgG antibodies. From the 228 control samples tested, 137 are positive for the presence of anti-*T. gondii* IgG antibodies (Table 1). Seroprevalence of anti-*T. gondii* IgG in the WOE and control groups is 72.8% (CI = 64.6–81.0%) and 60.1% (CI = 54.6–65.6%), respectively. Chi-square test with Yates’s correction was found to be 4.8181, showing that the difference in anti-*T. gondii* IgG seroprevalence between WOE and controls is statistically significant (*p* = 0.0282).

The association between the detection of anti-*T. gondii* IgG in the group of WOE and the variables (region, gender, age group, profession and number of years of work) was evaluated by binomial logistic regression (univariate) and multinomial logistic regression analysis (multivariate) (Table 2).

Being more than 50 years old (cOR 14.9; 95% CI 3.3–66.3; *p* < 0.001) was found to be associated with anti-*T. gondii* IgG seropositivity in binomial logistic regression (univariate analysis). Multivariate analysis also shows that the only risk factor for anti-*T. gondii* IgG seropositivity was the older age of the WOE (>50 years) with an aOR of 16.8 (95% CI 3.6–77.5; *p* < 0.001). None of the other variables concerning professional activity (slaughterhouse worker vs. butcher/veterinarian/farmer), gender (female vs. male), region (Centre vs. North) and years in practice (>16.5 vs. ≤16.5) are observed to be a significant risk factor for anti-*T. gondii* IgG seropositivity in WOE, by both univariate or multivariate analysis.

## 3. Discussion

In the present study, we report a significantly higher (*p* = 0.0282) anti-*T. gondii* IgG seroprevalence in WOE (75.8%; CI = 64.6–81.0%) compared with the general population (60.1%; CI = 54.6–65.6%). However, when ascertaining variables that could potentially be risk factors for higher seroprevalence of anti-*T. gondii* IgG, only age (>50 years old) was considered to be significant (*p* < 0.001).

The significant difference in anti-*T. gondii* IgG seroprevalence found in WOE and the general population (75.8% vs. 60.1%) was not surprising since other seroepidemiological surveys have reported similar results, with higher seroprevalences in those exposed to animals. A recent study from central India reported an anti-*T. gondii* seroprevalence of 48.9% for veterinarians and 48.4% for slaughterhouse workers against 6.6% in the control group [15]. This was also found to occur in Nigeria, where an anti-*T. gondii* IgG seroprevalence of 55.8% in WOE (slaughterhouse workers) vs. 32% of the general population was reported [16]. In Iran, butchers presented 48.8% anti-*T. gondii* IgG vs. 28.8% in the control group [12]. A systematic review performed in 2020 supports these findings, affirming that WOE had, in general, higher anti-*T. gondii* seroprevalence compared with general population. Despite this, evaluating risk factors frequently posed multiple difficulties mainly due to their complex relationship [7]. Although our results are consistent with various other studies, other authors have provided distinct conclusions. In a study from 2011, significant differences between WOE (slaughterhouse workers and butchers) and the control group were not found, showing no evidence that working with raw meat increased the probability of infection by *T. gondii* [10]. A few years later, the same team again presented data supporting lack of evidence for anti-*T. gondii* IgG association with WOE [13]. Other studies from 2008 and 2016 also reported no evidence for anti-*T. gondii* IgG differences between WOE and controls [19,20]. Of note, it was interesting to find a high anti-*T. gondii* seroprevalence in the controls of the present study (60.1%), which was higher than in the controls of other countries (6.6% to 32%) [12,15,16]. However, caution must be taken on this comparison as this might be the reflection of using distinctly aged groups or immunoassays with different sensitivities and specificities. A high seroprevalence in the Portuguese controls may be due to the important circulation of *T. gondii* in domestic animals and the consumption habits of the population [21].

When investigating risk factors for anti-*T. gondii* seropositivity in WOE, only age (>50 years old) shows a positive association, with an aOR of 16.8 (95% CI 3.6–77.5; *p* < 0.001). Hence, workers of more than 50 years of age are considered to have a 16.8-fold increased likelihood of having anti-*T. gondii* IgG than those less than 50 years old. Other studies reported age as a risk factor for the presence of anti-*T. gondii* IgG in WOE, namely in Japan, Malaysia, Iran, Finland and Poland, but all with a different cut-off age and focusing on different professions, from veterinarians to farmers [14,17,19,22,23]. This increased risk with age is not unexpected and is likely due to the longer period of exposure to risk factors, cumulatively occurring during the life-time exposure. Interestingly, years of practice did not appear to be a significant risk factor in the present study, as was in other reports [14].

In the present study, despite the inclusion of variables related to occupational exposure, several risk factors for the presence of anti-*T. gondii* IgG were not considered, such as those relating to consumption habits (raw vs. cooked meat, sheep/pig/chicken meat vs. beef) and hygiene at home (in the management of vegetables, water and cats), all of which are considered to be important [1].

To the best of our knowledge, this is the first epidemiological serosurvey searching for anti-*T. gondii* IgG in workers occupationally exposed to animals and their raw meat and viscera carried out in Portugal, showing that these WOE have an increased risk of infection with *T. gondii* compared with the general population controls. It is crucial that workers that are at continued risk of exposure re-enforce hygiene measures at work, such as not eating during labour time, and using protection equipment like gloves, masks and protective glasses [24]. Educating farmers about the existence of *T. gondii* and its importance, having a controlled and protected farm (strong biosecurity not allowing the presence of cats) and using clean water for crops are likely to have a positive influence in reducing the risk for *T. gondii* infection in animals and ultimately humans [1,4].

## 4. Materials and Methods

### 4.1. Sampling

For this study, archived (−80 °C) sera from a previous study were used [25]. WOE from the North and Centre of Portugal (*n* = 114) were selected, including 96 slaughterhouse workers (swine slaughterhouse), 5 butchers, 11 veterinarians and 2 farmers. Retrieved participants information included region (North or Centre of Portugal), gender (female or male), age (≤50 or >50 years old), professional activity (slaughterhouse worker or butcher/veterinarian/farmer) and number of years performing that job (≤16.5 or >16.5 years). The participant’s age ranged from 20 to 83 years, and they had been working with animals for 1 month up to 55 years. For levels dividing the variable “years of work” (≤16.5 or >16.5 years) and age (≤50 or >50 years old), the median of each distribution was identified and used. Descriptive data regarding selected WOE can be found in Table 3.

Of the control group, archived sera samples from anonymous volunteers (*n* = 228) matched with the WOE by region, gender and age were used in the proportion of two controls to each member of the WOE group. All procedures were conducted in accordance with the recommendations outlined in the Declaration of Helsinki and were approved by a national ethics board in Portugal (Comissão de Ética para a Saúde CHSJ; reference number: 99/2015).

### 4.2. Detection of Anti T. gondii Antibodies

All sera samples were individually tested for the presence of anti-*T. gondii* IgG antibodies using a commercial semiquantitative enzyme immunoassay (Toxoplasma ELISA IgG G1027, Edition 2018, Vircell, Granada, Spain). This test contains purified *T. gondii* antigen RH (ATCC 50174), and, according to the manufacturer it has a sensitivity of 98% (88–100%, confidence interval [CI] = 95%) and a specificity of 100% (89–100%, CI = 95%). All the procedures were performed according to the manufacturer’s instructions.

### 4.3. Data Analysis

Data processing was firstly performed using Microsoft Office 365 Excel. After organizing all the data, the statistical analysis was performed using IBM SPSS version 28.0.0.0 statistical software. A CI was established at 95%. A chi-square with Yate’s correction test for homogeneity of proportions was used to calculate significant differences in anti-*T. gondii* IgG seroprevalence between the WOE group and the control group. Binary and multinomial logistic regression analyses were carried out to determine which of the variables (region, gender, age, profession and years of work) were significantly (*p* < 0.05) associated with the detection of anti-*T. gondii* IgG among the WOE.

## Figures and Tables

**Table 1 pathogens-11-00603-t001:** Distribution of positive and negative anti-*T. gondii* IgG antibody results from the serum samples of the WOE (*n* = 114) and the control cases (*n* = 228).

	Positive No./% Anti-*T. gondii* IgG Antibody	Negative No./% Anti-*T. gondii* IgG Antibody	No. Total
WOE	83/72.8%	31/27.2%	114
Controls	137/60.1%	91/39.9%	228
Total	220/64.3%	122/35.7%	342

**Table 2 pathogens-11-00603-t002:** Univariate and multivariate analysis of risk factors for anti-*T. gondii* IgG seropositivity among Portuguese workers occupationally exposed to animals.

Variable	No. Positive (%)	Univariate Analysis cOR (95% CI)/*p* Value	Multivariate Analysis aOR (95% CI)/*p* Value
Professional activity			
Slaughterhouse worker	69 (71.9%)	Ref.	Ref.
Butcher/Veterinarian/Farmer	14 (77.8%)	1.4 (0.4–4.5)/0.607	1.9 (0.4–8.2)/0.406
Gender			
Female	24 (68.6%)	Ref.	Ref.
Male	59 (74.6%)	1.4 (0.6–3.2)/0.499	1.8 (0.7–5.1)/0.236
Age			
≤50 years	41 (58.6%)	Ref.	Ref.
>50 years	42 (95,5%)	14.9 (3.3–66.3)/<0.001	16.8 (3.6–77.5)/<0.001
Region			
North	42 (82.8%)	Ref.	Ref.
Centre	41 (73.2%)	1 (0.5–2.4)/0.924	0.8 (0.3–2.2)/0.630
Years in Practice			
≤16.5 years	41 (71.9%)	Ref.	Ref.
>16.5 years	42 (73.7%)	0.9 (0.4–2.1)/0.915	1.2 (0.5–3.1)/0.672

cOR: crude odds ratio, aOR: adjusted odds ratio, Ref.: variables reference level, CI: confidence interval.

**Table 3 pathogens-11-00603-t003:** Distribution of WOE by professional activity, gender, age, region and years of work.

Variable	No. Total (%)
Professional activity	
Slaughterhouse worker	96 (84.2%)
Butcher/Veterinarian/Farmer	18 (15.8%)
Gender	
Female	35 (30.7%)
Male	79 (69.3%)
Age	
≤50 years	70 (61.4%)
>50 years	44 (38.6%)
Region	
North	58 (50.9%)
Centre	56 (49.1%)
Years of Work	
≤16.5 years	57 (50%)
>16.5 years	57 (50%)

## Data Availability

Not applicable.

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
