# Peer review of "Prevalence of Toxoplasma gondii Antibodies in Individuals Occupationally Exposed to Livestock in Portugal"

_pathogens, 2022, doi:10.3390/pathogens11050603_

Round 1

Reviewer 1 Report

pathogens-1725620-peer-review-MCK minor comments line 41 – route Line 50-52 - incomplete sentence. Link 56-62 – improvise the grammar and make meaningful sentences. Line 63 – they Line 74 – Chi-square and not Qui-square Line 74 – the Yates correlation is expressed as 4.8181 and not as 4,8181 Results – line 80 - Describe the percentages in the table also Line 167 – Incomplete sentence. Line 177 – Sensitivity Line 182 – Microsoft Office 365 Excel Major comments Table 2. How and why were the age grouping was decided as ≤ 50 and ≥50? any scientific reason or justifications? Table 2. Years in practice - How and why was the years in practice grouping was decided as ≤ 16.5 and ≥16.5? any scientific reason? Line 135 – If WOE has higher risk of being infected with toxoplasmosis, as seen in the present and earlier studies, increase in years of practice by WOE should have increased the chance/ risk of toxoplasmosis. However, this was not the case in this study. What could be the possible reasons for such as observation? This observation kind of supports the observation by other scientists work in the references 10,13,19 and 21. What valid points does authors have justifying this statement? Line 173 – Has the authors determined the baseline titres among the healthy population for toxoplasmosis to use as a cut off titre? Or is there a baseline titre experimentally determined for the Portugal population by other researchers? The cut off titres given by the commercial diagnostic kit may not be suitable for all populations. Determining the endemic baseline titres helps in clearly defining the true cases vs subclinical or asymptomatic infection.

Author Response

# Reviewer 1:

Q1. line 41 – route

A1. We have performed as required.

Q2. Line 50-52 - incomplete sentence.

A2. We completed it.

Q3. Link 56-62 – improvise the grammar and make meaningful sentences.

A3. We have restructured the sentence.

Q4. Line 63 – they

A4. We have performed as required.

Q5. Line 74 – Chi-square and not Qui-square

A5. We have corrected as required.

Q6. Line 74 – the Yates correlation is expressed as 4.8181 and not as 4,8181

A6. Apologies, we have now corrected as required.

Q7. Results – line 80 - Describe the percentages in the table also

A7. We have performed as required.

Q8. Line 167 – Incomplete sentence.

A8. We completed it.

Q9. Line 177 – Sensitivity

A9. We have performed as required.

Q10. Line 182 – Microsoft Office 365 Excel

A10. We have performed as required.

Q11. Table 2. How and why were the age grouping was decided as ≤ 50 and ≥50? any scientific reason or justifications?

A11. We chose to use the median of the distribution to obtain the higher number possible in each category.

Q12. Table 2. Years in practice - How and why was the years in practice grouping was decided as ≤ 16.5 and ≥16.5? any scientific reason?

A12. For the same reason. We chose to use the median of the distribution to obtain the higher number possible in each category.

Q13. Line 135 – If WOE has higher risk of being infected with toxoplasmosis, as seen in the present and earlier studies, increase in years of practice by WOE should have increased the chance/ risk of toxoplasmosis. However, this was not the case in this study. What could be the possible reasons for such as observation? This observation kind of supports the observation by other scientists work in the references 10,13,19 and 21. What valid points does authors have justifying this statement?

A13. Yes, it would be expected that increase in years of practice would have increased the risk of toxoplasmosis. As suggested per the reviewers, we agree that more significantly than the risk factors associated with occupationally exposed work, are the ones related to the consumption habits at home as supported by the studies with references 10,13,19 and 21. Unfortunately our study did not evaluate further individual risk factors, particularly those associated to non-professional exposure.

Q14. Line 173 – Has the authors determined the baseline titres among the healthy population for toxoplasmosis to use as a cut off titre? Or is there a baseline titre experimentally determined for the Portugal population by other researchers? The cut off titres given by the commercial diagnostic kit may not be suitable for all populations. Determining the endemic baseline titres helps in clearly defining the true cases vs subclinical or asymptomatic infection.

A14. No data has been robustly drawn for the Portuguese population hence no adjustments outside of the manufacturers recommendation were made. Noteworthy, Vircell is a Spanish company and the assay was developed using sera from the neighboring country. Despite this, studies have used the same approach ( doi: 10.1080/00365540600978880. and https://doi.org/10.1186/s12876-018-0796-9 ) and a recent one has even found that it is highly correlated to immuno-PCR (https://doi.org/10.1007/s11686-022-00537-1)

Reviewer 2 Report

The manuscript “Prevalence of Toxoplasma gondii antibodies in individuals occupationally exposed to livestock in Portugal” by Almeida et al., provides a case-control study in which 114 workers occupationally exposed to animals or raw meat and viscera (WOE) [(pig slaughterhouse workers (96), butchers (5), veterinarians (11), and farmers (2)] and 228 archived sera from anonymous volunteers (in proportion two to one compared to WOE) were analyzed for anti-T. gondii IgG presence by a commercial ELISA. The authors observed significantly higher anti-T. gondii IgG occurrence in WOE (72.8%) compared to the general population (60.1%) which would be indication of an added risk for T. gondii infection in those exposed to animals or their meat and viscera. A multivariate analysis showed that WOE more than 50 years of age were more likely to be seropositive for anti-T. gondii IgG (aOR=16.8; (95% CI 3.6-77.5; p<0.001) than those with age less than 50 years.

The data are of interest, and it is the first case-control study on the prevalence of anti-T. gondii IgG in WOE in Portugal, although there are previous studies in other countries. The fact that is a case control is a positive point for the study.

The analysis of IgM antibodies against T. gondii in the sera samples would have been of interest to analyze recent contact with the parasite. Could the authors include those data?  

About the categorization of the variables, I would guess the reason to join the data of butchers (5), veterinarians (11), and farmers (2) for the statistical analysis was the low number of samples from some of those groups?

In the categorization of the variable “years of work” the median of each distribution was used so the proportion is 50% for each.  Why was the same not done for the age group? How was decided to separate the age variable in more or less than 50 years of age?

In the category “years of work” only 100 samples are included. Were data missing from the other samples for that variable?

Abstract: indicate that the sera analyzed were from archived samples.

Minor comments/corrections:

All scientific names need to be in cursive throughout the manuscript.

Line 41, “routes” not “routs”

Line 42. Delete the’ in “humans’

Line 44, add a coma after “immunosuppressed”

Lines 51 to 55, Rewrite the sentence, may be something like “A few epidemiological serosurveys tried to ascertain the risk for T. gondii infection in those in close contact with animals, their raw meat and viscera has grown, and this relationship [10,11]”.

Line 65. Change as “To increase the knowledge of this…”

Line 127. Add “the” before “risk factors for….”.

Line 130. Eliminate “aging” and indicates as” those less than 50 years old”

Table 1. Add the percentages of positivity and negativity for each group as well as the statistical significance among those groups in the table for more clarity.

In the discussion the fact that the controls show very high seroprevalence of T. gondii infection in Portugal should also be indicated.

Please correct Reference #22

Cursive needed in all the scientific names in reference 23.  

Author Response

# Reviewer 2:

Q1. The manuscript “Prevalence of Toxoplasma gondii antibodies in individuals occupationally exposed to livestock in Portugal” by Almeida et al., provides a case-control study in which 114 workers occupationally exposed to animals or raw meat and viscera (WOE) [(pig slaughterhouse workers (96), butchers (5), veterinarians (11), and farmers (2)] and 228 archived sera from anonymous volunteers (in proportion two to one compared to WOE) were analyzed for anti-T. gondii IgG presence by a commercial ELISA. The authors observed significantly higher anti-T. gondii IgG occurrence in WOE (72.8%) compared to the general population (60.1%) which would be indication of an added risk for T. gondii infection in those exposed to animals or their meat and viscera. A multivariate analysis showed that WOE more than 50 years of age were more likely to be seropositive for anti-T. gondii IgG (aOR=16.8; (95% CI 3.6-77.5; p<0.001) than those with age less than 50 years. The data are of interest, and it is the first case-control study on the prevalence of anti-T. gondii IgG in WOE in Portugal, although there are previous studies in other countries. The fact that is a case control is a positive point for the study.

A1. Thank you very much for your comments.

Q2. The analysis of IgM antibodies against T. gondii in the sera samples would have been of interest to analyze recent contact with the parasite. Could the authors include those data? 

A2. No. Unfortunately most samples have been exhausted (sera were also used for another study on HEV) and we cannot expand the research to other markers. Our apologies

Q3. About the categorization of the variables, I would guess the reason to join the data of butchers (5), veterinarians (11), and farmers (2) for the statistical analysis was the low number of samples from some of those groups?

A3. Yes, precisely. Apologies for not being clear

Q4. In the categorization of the variable “years of work” the median of each distribution was used so the proportion is 50% for each.  Why was the same not done for the age group? How was decided to separate the age variable in more or less than 50 years of age?

A4. Indeed we have tried to obtain the maximum number of samples per category. For this we determine the median, dividing the distribution in equal numbers, hence aiming towards the higher number of samples in both groups

Q5. In the category “years of work” only 100 samples are included. Were data missing from the other samples for that variable?

A5. The reviewer is absolutely correct! This was a Typo. We have now corrected this mistake!

Q6. Abstract: indicate that the sera analyzed were from archived samples.

A6. We have performed as required.

Q7. All scientific names need to be in cursive throughout the manuscript.

A7. We have performed as required.

Q8. Line 41, “routes” not “routs”

A8. We have performed as required.

Q9. Line 42. Delete the’ in “humans’

A9. We have performed as required.

Q10. Line 44, add a coma after “immunosuppressed”

A10. We have performed as required.

Q11. Lines 51 to 55, Rewrite the sentence, may be something like “A few epidemiological serosurveys tried to ascertain the risk for T. gondii infection in those in close contact with animals, their raw meat and viscera has grown, and this relationship [10,11]”.

A11. We have rewritten the sentence.

Q12. Line 65. Change as “To increase the knowledge of this…”

A12. We have performed as required.

Q13. Line 127. Add “the” before “risk factors for….”.

A13. We have performed as required.

Q14. Line 130. Eliminate “aging” and indicates as” those less than 50 years old”

A14. We have performed as required.

Q15. Table 1. Add the percentages of positivity and negativity for each group as well as the statistical significance among those groups in the table for more clarity.

A15. We have performed as required

Q16. In the discussion the fact that the controls show very high seroprevalence of T. gondii infection in Portugal should also be indicated.

A16.  We have performed as required in between lines 131 and 134. Even though we didn’t explore this subject much, as we think it also requires a study for itself, including risk factors for the general population. Probably one of our next steps of investigation!!

Q17. Please correct Reference #22. Cursive needed in all the scientific names in reference 23. 

A17. We have performed as required.